PREPARED FOR SUBMISSION TO JHEP

# Warped AdS$_3$ black hole thermodynamics and the charged Cardy formula

**Kiril Hristov**[a,b] **and Riccardo Giordana Pozzi**[c,d]

[a] *Faculty of Physics, Sofia University, J. Bourchier Blvd. 5, 1164 Sofia, Bulgaria*

[b] *INRNE, Bulgarian Academy of Sciences, Tsarigradsko Chaussee 72, 1784 Sofia, Bulgaria*

[c] *Dipartimento di Scienze Fisiche, Informatiche e Matematiche,*
*Università di Modena e Reggio Emilia, via G. Campi 213/A, 41125 Modena, Italy*

[d] *INFN Sezione di Bologna, via Irnerio 46, 40126 Bologna, Italy*

*E-mail:* khristov@phys.uni-sofia.bg, riccardo.pozzi@unimore.it

ABSTRACT: We revisit the thermodynamic properties of black holes with warped AdS$_3$ asymptotics in topologically massive gravity. The holographically dual theory, often referred to as warped CFT$_2$, exhibits a single copy of the Virasoro-Kac-Moody algebra. Consequently, the asymptotic density of states in the right-moving sector is described by the charged Cardy formula that we review in detail. By defining specific linear combinations of the gravitational thermodynamic potentials, we are able to present the gravitational left- and right-moving on-shell actions directly in the grand-canonical ensemble. This allows us to demonstrate that, apart from the charged Cardy behavior in the right-moving sector, the dual field theory exhibits an additional *frozen* left-moving state giving rise to non-zero entropy contribution. Our findings elucidate the nature of warped CFTs and reveal subtle yet fundamental differences from the existing literature.

## 1 Introduction and main results

The *charged* or *flavored* Cardy formula, a generalization of the Cardy formula for (either the left- or the right-moving sector of) a CFT$_2$ with an additional $U(1)$ symmetry at level $k$, provides the asymptotic density of states on a torus at energy level $n$ within a fixed $U(1)$ charge sector $q$,

$$\log \rho(n,q) \approx 2\pi \sqrt{\frac{c}{6}\left(n - \frac{c}{24} - \frac{q^2}{2\,k}\right)}, \tag{1.1}$$

where $c$ is the central charge. This formula, presented here in the microcanonical ensemble of fixed $n, q$ and varying conjugate variables $\tau, \mu$, can be extended to include multiple $U(1)$ levels. Remarkably, it has been derived and (seemingly independently) rederived many times in the physics literature due to its numerous applications in microscopic entropy counting, which reproduce the gravitational Bekenstein-Hawking formula. An incomplete list of references and applications to D-brane constructions is given by [1–5], see also [6] for a more recent exposition and application in novel AdS/CFT settings.

In the context of the present work, a version of the charged Cardy formula appears to have been once again independently derived via modularity in [7] in the description of the so-called *warped* CFTs (WCFTs). [1] These are proposed as holographic duals to the *warped* AdS$_3$ geometry that breaks some of the asymptotic AdS$_3$ symmetries, in particular exhibiting only a single copy of the Virasoro-Kac-Moody algebra. However, despite various efforts, see [8–10], there is no universal agreement on the microscopic definition of warped CFTs. In our desire to understand them better we therefore turn our attention to the gravitational systems with some new observations.

Warped AdS$_3$, or WAdS$_3$ [11], appears as a solution of several different three-dimensional gravitational theories with a cosmological constant, united by the appearance of a massive degree of freedom. In the present work we are going to use topologically massive gravity (TMG), [12, 13], whose solutions have been discussed in [14–19] and references thereof. However, WAdS$_3$ shows up in other theories with massive vectors or higher derivative terms, see [20–24]. The characteristic feature of WAdS$_3$ due to the additional warp factor with respect to the standard AdS$_3$ metric, is that the asymptotic symmetry group is given by $SL(2,\mathbb{R}) \times U(1)$, down from the full set of AdS$_3$ isometries, $SL(2,\mathbb{R})_l \times SL(2,\mathbb{R})_r$. In Euclidean signature and in the presence of suitable boundary conditions, see [25, 26], the asymptotic WAdS$_3$ symmetries further enhance to the full Virasoro-Kac-Moody algebra that we review in the next section.

Based on symmetries alone, one heuristically expects that physics in WAdS$_3$ and the dual WCFT$_2$ is going to be a chiral version of the AdS$_3$/CFT$_2$ case, and this is indeed true to a large extent with some interesting surprises. The thermodynamics of different black holes in the gravity theory has been considered in various references, starting from [11] and numerous references thereof, see e.g. [27] for a recent review of the state of the art. Based on the number of times the gravitational system has been reviewed and revisited, it seems fair to assess that the true nature of the WCFT and its exact relation to a usual CFT has not been completely established. In this context the results we present here give a very precise and novel suggestion. The WCFT appears to behave as a standard CFT in its right-moving sector, whereas the left-moving sector has no dynamics and consists of a single state (in the grand-canonical ensemble) with vanishing charges. In the microcanonical description this left-moving state corresponds to a multiparticle system with non-vanishing constant $U(1)$ chemical potential, which is in a sense *frozen*. At constant $U(1)$ charge the left-moving sector thus contributes with non-zero entropy, in addition to the right-moving charged Cardy behavior of the density of states.

In the bulk of our gravitational analysis we revisit the thermodynamics of warped AdS black holes. [2] The new element here is the use of the so-called *natural* variables, first introduced in [28] within a 4d Einstein-Maxwell framework and later generalized to

---

[1]Note that an additional imaginary phase has been included in [7], which typically is ignored as it is suppressed asymptotically. This leads to the proposition of allowing an imaginary asymptotic charge, which we do not need here. The precise relation of the present work with [7] is explained in due course.

[2]We follow the convention of [27] on the naming of black hole solutions. For completion and additional clarity we also consider the warped BTZ black holes and the warped dS black holes in the appendices.

various theories and black hole examples in [29, 30]. This approach is based on defining a set of chemical potentials that satisfy the first law of thermodynamics at each black hole horizon separately [31], see also [32–34]. One can thus take linear combinations of the corresponding variables while preserving the conservation law, and for the present case with two different horizons [28] defined corresponding left- and right-moving entropies, temperatures, and, ultimately, left- and right-moving on-shell actions. These on-shell actions, $I_{l,r}$, exhibit a remarkable empirical property: they become *significantly* simpler functions of the corresponding chemical potentials.

At first glance, this construction seems to double the number of chemical potentials while maintaining the same number of asymptotic charges at odds with microscopic expectations. However, as demonstrated in [30], a remarkable property holds in 3d systems: exactly half of the left- and right-moving potentials are constants and are thus not independent. Consequently, these algorithmically defined natural variables automatically match the dual CFT's left- and right-moving (or holomorphic and anti-holomorphic) variables. The introduction of the inner horizon, therefore, is a simple trick to algorithmically define holographically suitable variables without actually doubling their number. This property holds for all black holes in warped (A)dS considered here, guiding us toward understanding the appropriate variables on the microscopic side as well.

Using these natural variables, we uncover the following thermodynamic behavior of the warped AdS$_3$ black holes, which naturally leads us back to the charged Cardy formula. The asymptotic charges, corresponding to the commuting part of the asymptotic symmetries, are the mass $M$ and angular momentum $J$, with their respective conjugate variables being the inverse temperature $\beta$ and angular velocity $\Omega$. By splitting into left- and right-moving variables, we define $\beta_l, \omega_l$ and $\beta_r, \omega_r$, which are conjugate to the same conserved $M$ and $J$, respectively. [3] It turns out that the left-moving sector vanishes identically in the grand-canonical ensemble, where $\beta_l$ is constant and $\omega_l = 0$. Therefore, the thermodynamics of the warped black hole is entirely governed by the right-moving sector,

$$I_l = 0 \ , \qquad I_r = -\frac{\pi}{12\,\omega_r}\left(c_r + \frac{3\,k_r}{\pi^2}\,\beta_r^2\right) \ , \qquad (1.2)$$

where the right-moving central charge $c_r$ and $U(1)$-level $k_r$ are determined by the theory's coupling constants (in our case, TMG). The expression for the right-moving action $I_r$ is essentially the grand-canonical version of the charged Cardy formula, derived from modular invariance as detailed in the next section. Returning to the microcanonical ensemble of fixed asymptotic charges, this yields the right-moving entropy,

$$S_r(J, M) = 2\pi\,\sqrt{\frac{c_r}{6}\left(-\frac{J}{2\pi} - \frac{M^2}{2\,k_r}\right)} \ , \qquad (1.3)$$

which aligns with the standard CFT expectation from (1.1). Interestingly, in the warped black holes, the angular momentum takes the place of the holographic energy, while the

---

[3]See the main text for the proper definition and difference between the notations for $\Omega$ and $\omega_{l,r}$, which is not just notational.

mass corresponds to the additional $U(1)$ charge, as dictated by the asymptotic symmetry algebra. See Eq. (4.5) for the precise mapping.

There is, however, one surprise in the microcanonical ensemble that distinguishes the behavior of warped CFTs from ordinary ones. In the left-moving sector, a unit partition function (corresponding to $I_l = 0$) typically signifies a BPS vacuum at vanishing temperature and a respective vanishing entropy contribution. This is not the case here. Although the left-moving on-shell action vanishes, there is a non-zero contribution to the entropy in the microcanonical ensemble due to the constant non-vanishing inverse temperature $\beta_l$. Consequently, the total entropy of the gravitational system is given by

$$ S = S_l + S_r = \beta_l\, M + 2\pi \sqrt{ \frac{c_r}{6} \left( -\frac{J}{2\pi} - \frac{M^2}{2\,k_r} \right) } \,, \tag{1.4} $$

suggesting an interesting dual picture that aligns with the analysis in [26] finding a *crossover* zero-mode of the $U(1)$ current on the left-moving side. [4] We interpret the dual WCFT as a combination of an ordinary right-moving conformal sector and a non-dynamic, or *frozen*, left-moving sector at constant $U(1)$ chemical potential in the microcanonical ensemble.

Note that our results rather subtly differ from the field theory match suggested in [7], which includes an additional imaginary phase for the right-movers and an imaginary vacuum expectation value of the $U(1)$ current, but ignores the possibility of a frozen left-moving state. This imaginary vev is a sign of a non-unitary theory, which we do not find to be needed here. This difference in our analysis has therefore a potentially fundamental importance as we do not find any obstructions for WCFTs to be unitary. We provide a more detailed account of this distinction in the next section, where we discuss the asymptotic Virasoro-Kac-Moody algebra.

The rest of the paper is organized as follows. In Section 2, we detail the asymptotic density of states resulting from the modular properties of the partition function and explain our understanding of the warped CFT. In Section 3, we present the gravitational theory and the black hole solutions of interest, analyzing their thermodynamics in terms of the newly defined potentials. Section 4 describes the holographic match between the two sides and proposes the form of the quantum entropy of the system based on modularity, concluding the main body of this work.

In the appendices, we discuss other classes of black holes within TMG that we also describe holographically. Appendix A examines the so-called warped BTZ black holes, showing that they correspond to an ordinary $CFT_2$ with different left- and right-moving central charges, $c_l \neq c_r$, due to the Chern-Simons term in TMG. In Appendix B, we flip the sign of the cosmological constant and discuss warped $dS_3$ black holes, highlighting their close similarities with the warped AdS case. The warped CFT discussion in the main text formally applies to the dS solutions as well, opening interesting possibilities.

---

[4] Crossover Kac-Moody currents have been observed also in standard $AdS_3/CFT_2$ settings such as the MSW black holes, [2], and 5d black strings dual to $\mathcal{N} = 4$ SYM, [6]. The characteristic feature that is true also in the present case is the negative Kac-Moody level, $k_r$, c.f. (3.5).

## 2  Charged Cardy formula and warped CFT

Here we focus on the field theory, at first reviewing well known results. We exploit the asymptotic symmetries of the gravity backgrounds to constrain the field theory partition function based on modularity. We first discuss a standard $\text{CFT}_2$, including the presence of an extra $U(1)$ symmetry, before turning to the warped case.

### 2.1  Pure Cardy formula

Let us start with a standard $\text{CFT}_2$, for the moment without any additional symmetries. In 2d we nevertheless have an infinite set of conserved charges that enhance the usual conformal group to two copies of the Virasoro algebra, with generators $L_n$ and $\bar{L}_n$, respectively called left- and right-moving.

The commutation relations are given by

$$[L_n, L_m] = (n - m) L_{n+m} + \frac{c_l}{12} (n^3 - n) \delta_{n+m} \ , \tag{2.1}$$

together with

$$[\bar{L}_n, \bar{L}_m] = (n - m) \bar{L}_{n+m} + \frac{c_r}{12} (n^3 - n) \delta_{n+m} \ , \tag{2.2}$$

and mixed generators vanish.

We define the theory on a torus with a complex structure parameter $\tau$, such that the partition function is a modular form, which (up to non-perturbative contributions that we ignore) factorizes into

$$Z_{\text{CFT}_2}(\tau, \bar{\tau}) = Z_l(\tau) \, Z_r(\bar{\tau}) \ , \tag{2.3}$$

with the holomorphic part defined as a trace over states of various energies

$$Z_l(\tau) := \text{Tr}\left[ e^{2\pi i \tau (L_0 - c_l/24)} \right] \ , \tag{2.4}$$

and the anti-holomorphic part defined analogously in terms of $\bar{\tau}$.

Individually, both the holomorphic and the anti-holomorphic parts enjoy the modular transformation property

$$Z_l(\frac{a\tau + b}{c\tau + d}) \sim Z_l(\tau) \tag{2.5}$$

and likewise for $Z_r(\bar{\tau})$, valid in the limit $\tau \to 0$ of interest from a dual gravity point of view. Neglecting subleading terms, we find

$$Z_l(\tau) = Z_l(-1/\tau) \sim e^{\frac{i\pi}{12\tau} c_l} \ , \tag{2.6}$$

where $Z_l(-1/\tau)$ is dominated by the vacuum, $L_0 = 0$. We thus arrive at the leading approximation,

$$\log Z_l(\tau) \approx \frac{i\pi}{12\tau} c_l \ , \qquad \log Z_r(\bar{\tau}) \approx \frac{i\pi}{12\bar{\tau}} c_r \ . \tag{2.7}$$

If we want to convert this to the microcanonical ensemble, we need to perform the corresponding Laplace transform,

$$\rho(n_l, n_r) = \int_{\mathcal{C}} d\tau \, Z_l(\tau) \, e^{-2\pi i \tau \left(n_l - \frac{c_l}{24}\right)} \int_{\mathcal{C}} d\bar{\tau} \, Z_r(\bar{\tau}) \, e^{-2\pi i \bar{\tau} \left(n_r - \frac{c_r}{24}\right)} \ , \tag{2.8}$$

leading to the saddle-point approximation for the asymptotic density of states

$$\log \rho(n_l, n_r) \approx 2\pi \sqrt{\frac{c_l}{6}\left(n_l - \frac{c_l}{24}\right)} + 2\pi \sqrt{\frac{c_r}{6}\left(n_r - \frac{c_r}{24}\right)} \ , \qquad (2.9)$$

where we have relabeled the fixed energy levels, $L_0 = n_l$ and $\bar{L}_0 = n_r$, as standardly done in literature.

## 2.2 Adding $U(1)$ currents

Now we add more symmetries to the CFT$_2$ by coupling the theory to additional $U(1)$ currents, which form an infinite Kac-Moody algebra. Note that we are allowed to independently add any number of $U(1)$ symmetries to the left- and right-moving Virasoro algebras. For simplicity, we look at the symmetric choice of having (independent) single $U(1)$ currents in both sectors, but the generalization is straightforward, see [6]. We introduce left-moving $Q$ and right-moving $\bar{Q}$, which in analogy with the Virasoro algebra get enhanced to an infinite number of operators in the Virasoro-Kac-Moody alegbras,

$$[L_n, L_m] = (n - m) L_{n+m} + \frac{c_l}{12} (n^3 - n) \delta_{n+m} \ ,$$
$$[Q_n, Q_m] = \frac{k_l}{2} \delta_{n+m} \ , \qquad\qquad\qquad\qquad (2.10)$$
$$[L_n, Q_m] = -m \, Q_{n+m} \ ,$$

together with

$$[\bar{L}_n, \bar{L}_m] = (n - m) \bar{L}_{n+m} + \frac{c_r}{12} (n^3 - n) \delta_{n+m} \ ,$$
$$[\bar{Q}_n, \bar{Q}_m] = \frac{k_r}{2} \delta_{n+m} \ , \qquad\qquad\qquad\qquad (2.11)$$
$$[\bar{L}_n, \bar{Q}_m] = -m \, \bar{Q}_{n+m} \ .$$

Note again that, just like the central charges $c_l$ and $c_r$, the $U(1)$ levels $k_l$ and $k_r$ are a priori independent. [5]

The partition function again factorizes

$$Z_{\mathrm{CFT}_2}(\tau, \mu, \bar{\tau}, \bar{\mu}) = Z_l(\tau, \mu) \, Z_r(\bar{\tau}, \bar{\mu}) \ , \qquad (2.12)$$

with $\mu$ and $\bar{\mu}$ the chemical potentials for $Q$ and $\bar{Q}$, respectively. The holomorphic part of the partition function is now defined via

$$Z_l(\tau, \mu) := \mathrm{Tr}\left[ e^{2\pi i \tau (L_0 - c_l/24) + i \mu Q_0} \right] \ , \qquad (2.13)$$

and likewise for the anti-holomorphic part in terms of $\bar{\tau}, \bar{\mu}$.

---

[5]In the presence of higher amounts of supersymmetry, see e.g. [4], the $U(1)$ current can get enhanced to a non-abelian group and the corresponding level will be proportional to the central charge. This is not needed for the present purposes.

This time the modular properties of the partition function involve also $\mu$, see e.g. [35][Sec. 2] for detailed derivation,

$$Z_l(\frac{a\tau+b}{c\tau+d}, \frac{\mu}{c\tau+d}) \sim \exp\left(\frac{ik_l}{4\pi}\frac{c\mu^2}{c\tau+d}\right) Z_l(\tau,\mu) , \tag{2.14}$$

and therefore

$$Z_l(\tau,\mu) = e^{-\frac{i\mu^2}{4\pi\tau}k_l} Z_l(-1/\tau, -\mu/\tau) \sim e^{\frac{i\pi}{12\tau}\left(c_l - \frac{3k_l}{\pi^2}\mu^2\right)} , \tag{2.15}$$

again in the limit $\tau \to 0$ where $Z_l(-1/\tau, -\mu/\tau)$ is dominated by the vacuum, $L_0 = 0$. Putting it together,

$$\log Z_l(\tau,\mu) \approx \frac{i\pi}{12\tau}\left(c_l - \frac{3k_l}{\pi^2}\mu^2\right) , \qquad \log Z_r(\bar\tau,\bar\mu) \approx \frac{i\pi}{12\bar\tau}\left(c_r - \frac{3k_r}{\pi^2}\bar\mu^2\right) . \tag{2.16}$$

Similarly, in the microcanonical ensemble we have,

$$\rho(n_l, q_l, n_r, q_r) = \int_{\mathcal{C}} \mathrm{d}\tau\mathrm{d}\mu\, Z_l(\tau,\mu)\, e^{-2\pi i\tau\left(n_l - \frac{c_l}{24}\right) - i\mu q_l} \int_{\mathcal{C}} \mathrm{d}\bar\tau\mathrm{d}\bar\mu\, Z_r(\bar\tau,\bar\mu)\, e^{-2\pi i\bar\tau\left(n_r - \frac{c_r}{24}\right) - i\bar\mu q_r} , \tag{2.17}$$

leading to

$$\log \rho(n_l, q_l, n_r, q_r) \approx 2\pi\sqrt{\frac{c_l}{6}\left(n_l - \frac{c_l}{24} - \frac{q_l^2}{2\,k_l}\right)} + 2\pi\sqrt{\frac{c_r}{6}\left(n_r - \frac{c_r}{24} - \frac{q_r^2}{2\,k_r}\right)} , \tag{2.18}$$

where we have relabeled the fixed $U(1)$ levels, $Q_0 = q_l$ and $\bar Q_0 = q_r$, for notational consistency.

## 2.3 Warped CFT$_2$

Given the above standard results, we can discuss more carefully how a warped CFT$_2$ differs from an ordinary CFT$_2$, based on the explicit gravity duals.

As will be described in the next section, one can show that warped AdS$_3$ preserves exactly one copy of the Virasoro-Kac-Moody algebra, which due to the conventions we follow is going to be in the right-moving sector, [6]

$$[\bar L_n, \bar L_m] = (n-m)\bar L_{n+m} + \frac{c_r}{12}(n^3 - n)\delta_{n+m} ,$$

$$[\bar Q_n, \bar Q_m] = \frac{k_r}{2}\delta_{n+m} , \tag{2.19}$$

$$[\bar L_n, \bar Q_m] = -m\bar Q_{n+m} .$$

We already noticed that the contributions of the two sectors factorize, such that the previous calculations can be applied verbatim to the right-moving sector. Defining

$$Z_r(\bar\tau,\bar\mu) := \mathrm{Tr}\left[e^{2\pi i\bar\tau(\bar L_0 - c_r/24) + i\bar\mu\bar Q_0}\right] , \tag{2.20}$$

---

[6]Note that in the original asymptotic symmetries of the holographically dual backgrounds, the additional current is actually non-compact as it relates to time translations. In order to discuss black hole thermodynamics, we perform Wick rotation and periodic time identification such that we are back to the compact $U(1)$ case.

we can immediately derive its leading expression from the previous subsection (see below).

On the other hand, based on the gravitational description and asymptotic symmetries we are lead to conclude that the left-moving sector of a warped CFT is not entirely absent even if there is no Virasoro-Kac-Moody algebra associated with it. The $U(1)$ current discussed above is in fact a *crossover* current that has a left-moving zero-mode, $Q_0 = \bar{Q}_0$, see [10, 26]. We thus find, based on our gravity analysis, that there exists a single left-moving state, invariant under the $U(1)$ symmetry,

$$Q_0 \left| \underline{\mu} \right\rangle = 0 \ , \tag{2.21}$$

i.e. it has vanishing charge, $Q_0 = 0$. We have no evidence for any other symmetries acting on the left-moving sector. We have labeled this state according to the holographic prediction that it exhibits a non-vanishing $U(1)$ chemical potential, $\underline{\mu}$, which will be important later. The left-moving partition function is therefore trivial, but importantly *not* absent,

$$Z_l(\mu) := \text{Tr} \left[ e^{i\mu Q_0} \right] = \left\langle \underline{\mu} \right| e^{i\mu Q_0} \left| \underline{\mu} \right\rangle = 1 \ . \tag{2.22}$$

Putting the two sectors together, we find at leading order

$$\log Z_{\text{WCFT}}(\bar{\tau}, \bar{\mu}) := \log Z_l(\mu) + \log Z_r(\bar{\tau}, \bar{\mu}) \approx \frac{i\,\pi}{12\,\bar{\tau}} \left( c_r - \frac{3 k_r}{\pi^2} \bar{\mu}^2 \right) \ . \tag{2.23}$$

This agrees precisely with the grand-canonical free energy of the black hole solutions in WAdS$_3$, as we show in due course. We caution the reader that the symmetries in the WCFT are actually rotated in relation to the gravitational dual, c.f. (4.2). What we typically call black hole temperature relates to the chemical potential $\bar{\mu}$, while the angular velocity relates to $\bar{\tau}$, which is usually reversed in standard holographic examples, see e.g. [6]. At the level of the symmetries this relation is simple to understand, since the rotations in the black hole language are actually part of the $SL(2,\mathbb{R})$ group, while the time-translations are the additional $U(1)$.

At this point it is useful to compare our results with those in [7], where the discussion starts directly from the WCFT algebra, (2.19), without referring to the charged Cardy formula of a standard CFT$_2$. The same formula, (2.23), was rederived using the modular properties that we reviewed, with the important addition of a purely imaginary phase, c.f. Eq. (44) in [7]. In the asymptotic evaluation of the partition function, (2.14)-(2.15), we have neglected such terms as they are invisible holographically. The absence of this term on the gravity side can be seen by a direct comparison between the two sides already in the grand-canonical ensemble, performed in section 4. Such a comparison is novel as it is based on the natural thermodynamic variables on the gravitational side.

We should however note that the extra term in [7], discussed above, has likely been introduced due to an additional subtlety for the WCFT in the microcanonical ensemble (fixed $n_r, q_r$). [7] As suggested by our gravitational analysis, the left-moving sector appears to be at a constant *non-vanishing* temperature, translating into a frozen value for $\mu$ on the field theory side. In the microcanonical ensemble the state $\left| \underline{\mu} \right\rangle$ discussed above can

---

[7]The analysis in [7] compares the gravity and field theory sides only in the microcanonical ensemble.

be reinterpreted as a multiparticle state, whose chemical potential $\underline{\mu} \neq 0$ does not vary with respect to the extensive quantity $Q_0$. Thus we observe an interesting phenomenon: adding fixed $U(1)$ charge $Q_0$ on this frozen multiparticle state results in an additional entropy contribution to the full density of states of the system. In order to account for this subtlety, an imaginary vacuum expectation value for the $U(1)$ charge $\bar{Q}_0$ is turned on in [7], which is a sign of a non-unitary theory. Our results here suggest that this is not needed, and therefore still allow the WCFT to be unitary.

In line with our frozen chemical potential interpretation, we thus find the microcanonical density of states for a warped CFT to be

$$\rho_{\mathrm{WCFT}}(\underline{\mu}; n_r, q_r) := e^{-i\underline{\mu}q_r} \int_{\mathcal{C}} \mathrm{d}\bar{\tau}\mathrm{d}\bar{\mu}\, Z_r(\bar{\tau}, \bar{\mu})\, e^{-2\pi i \bar{\tau}\left(n_r - \frac{c_r}{24}\right) - i\bar{\mu}q_r}\,, \qquad (2.24)$$

where we have identified both the left-moving $Q_0$ and the right-moving $\bar{Q}_0$ with the same fixed charge $q_r$ as they correspond to the same symmetry. At a leading order we therefore recover the charged Cardy formula, shifted by a constant,

$$\log \rho \approx -i\underline{\mu}\, q_r + 2\pi \sqrt{\frac{c_r}{6}\left(n_r - \frac{c_r}{24} - \frac{q_r^2}{2\,k_r}\right)}\,. \qquad (2.25)$$

We should stress that the additional left-moving term here is motivated from the dual gravitational picture, as we have no microscopic understanding of the value of $\underline{\mu}$. However, empirically we will find that $\underline{\mu}$ is related to the right-moving $U(1)$ level, $k_r$, see (4.4). Such a relation was already observed in similar settings in [8, 26] based on the asymptotic symmetries, giving additional credibility of our results. We thus reach our main insight in the nature of a warped CFT, as already anticipated in the introduction. A WCFT behaves fully as a CFT in the right-moving sector, while exhibiting a frozen non-dynamic left-moving sector at a constant $U(1)$ chemical potential.

Finally, notice that the additional left-moving term is in fact crucial in distinguishing a WCFT from an ordinary CFT that saturates the extremal/supersymmetric limit. As explained in a similar setting in [30] (and many previous references) the extremal limit corresponds to $\tau \to \infty$, such that the left-moving sector is again trivial in the grand-canonical ensemble, but in this case it does not contribute to the density of states in the microcanonical ensemble.

## 3 Thermodynamics of warped AdS$_3$ black holes

We consider the action of topologically massive gravity, [12, 13],

$$I_{\mathrm{TMG}} = \frac{1}{16\pi G_3}\left[\int \mathrm{d}^3 x \sqrt{-g}\,(R - 2\Lambda)\right] + \frac{1}{2\,\mu}\, I_{\mathrm{CS}}\,, \qquad (3.1)$$

with

$$I_{\mathrm{CS}} = \frac{1}{16\pi G_3}\left[\int \mathrm{d}^3 x \sqrt{-g}\,\varepsilon^{\lambda\mu\nu}\,\Gamma^{\alpha}_{\lambda\sigma}\,(\partial_\mu \Gamma^{\sigma}_{\alpha\nu} - \tfrac{2}{3}\Gamma^{\sigma}_{\mu\tau}\Gamma^{\tau}_{\nu\alpha})\right]\,, \qquad (3.2)$$

where $G_3$ is the 3d Newton constant, $\Lambda = \pm 1/l^2$ with the $+$ sign used for dS solutions and $-$ for AdS, and $\mu$ taken as an arbitrary positive Chern-Simons coupling constant that has a unit of mass. [8] Conventionally we define

$$\nu := \frac{\mu l}{3} \ ,$$

such that the resulting dimensionless parameter $\nu$ is taken as a positive number that is typically used in categorizing the solutions. Here we will be looking at the case of negative cosmological constant.

## 3.1 Warped solutions of TMG

There are two main classes of solutions in TMG with $\Lambda < 0$: locally AdS$_3$ backgrounds, with vanishing cotton tensor, and locally warped AdS$_3$ (WAdS) backgrounds with non-vanishing cotton tensor, see [36] and references therein for a detailed classification. The important distinction of the latter solutions is that they explicitly depend on the dimensionless parameter $\nu$, and can be understood as line fibrations over (either Lorentzian or Euclidean) AdS$_2$, such that the symmetries $U(1) \times SL(2,\mathbb{R})$ are manifest but there is no enhancement to the full AdS$_3$ symmetry algebra, $SO(2,2)$. Depending on the signature of the fibration, the WAdS vacua can be either spacelike (with Lorentzian AdS$_2$ base), timelike (with Euclidean AdS$_2$ base), or null, and in addition the first two categories allow for two distinct branches called stretched ($\nu^2 > 1$) and squashed ($\nu^2 < 1$).

Note that the above distinction between spacelike and timelike warped AdS$_3$ is actually misleading for our purposes. Both in field theory (when using the infinite dimensional Virasoro algebra) and in gravity (when discussing finite temperature black holes with periodic time) we are forced to work in Euclidean signature. This means that we must perform a Wick rotation of TMG and consider Euclidean warped AdS$_3$ (E-WAdS), both the signature of the fibration and signature of the base are positive definite. We thus look at the unique choice for spacelike E-WAdS,

$$\mathrm{d}s^2_{\text{E-WAdS}} = \frac{l^2}{\nu^2 + 3} \left[ \cosh^2 \sigma \, \mathrm{d}\tau^2 + \mathrm{d}\sigma^2 + (\mathrm{d}u + \sinh \sigma \, \mathrm{d}\tau)^2 \right] \ , \tag{3.3}$$

where both $\tau$ and $u$ are compact coordinates. It is important to track down the symmetries of the metric, which are carefully analysed in [11][App. A]: the $\partial_\tau$ Killing vector is part of the $SL(2,\mathbb{R})$, while $\partial_u$ gives rise to the additional $U(1)$ isometry.

From a Lorentzian perspective, regular black hole spacetimes without closed timelike curves (CTCs) exist only in the spacelike stretched branch as shown in [11]. On the other hand the timelike branch is interesting as it admits supersymmetry when embedded in higher derivative supergravity theories, [37, 38], and has an integrable structure inside string theory, [39]. Given that the distinction disappears in Euclidean signature, we are considering both branches at the same time, such that our analysis is insensitive

---

[8]The Chern-Simons coupling $\mu$ in this section has no relation to the $U(1)$ chemical potential of the previous section. There should be no confusion in what follows as we swiftly switch to the parameter $\nu$ in the gravitational analysis.

to any causality issues. On the other hand it would be interesting to understand if the (Euclideanized) black holes we discuss next could exhibit a supersymmetric limit, which we leave here as an open question.

Let us finally turn to the main subject of interest, the quotients of the above metric that have an interpretation of black holes in analogy to the BTZ black holes, [40, 41]. Spacelike WAdS black holes are more conveniently described by the metric, [9]

$$\frac{ds^2}{l^2} = dt^2 + \frac{dr^2}{(\nu^2 + 3)(r - r_+)(r - r_-)} + \left(2\nu r - \sqrt{r_+ r_-(\nu^2 + 3)}\right) dt d\theta$$
$$+ \frac{r}{4}\left(3(\nu^2 - 1)r + (\nu^2 + 3)(r_+ + r_-) - 4\nu\sqrt{r_+ r_-(\nu^2 + 3)}\right) d\theta^2 \quad (3.4)$$

where, as usual, $r_+$ and $r_-$ are the locations for the outer and inner black hole horizons respectively. Importantly, the relation between the black hole coordinates in this form of the metric and the direct quotients of E-WAdS in the form above is carefully recorded in [11], with $t$ being most directly related to $u$, and $\theta$ to $\tau$. This means that the (compactified) "time translations" of the black holes are the additional $U(1)$ current on the field theory, while rotations along $\theta$ are part of the $SL(2, \mathbb{R})$ group.

It was shown in [25] that for the background above it is possible to impose asymptotic boundary conditions that enhance the symmetry to the full Virasoro-Kac-Moody algebra discussed in the previous section, either in the left-moving or in the right-moving sector (but not both). In our conventions we pick the right-moving choice, where the algebra is defined by the central charge and $U(1)$ level,

$$c_r = \frac{2\pi l(5\nu^2 + 3)}{\nu(\nu^2 + 3) G_3} , \qquad k_r = -\frac{\pi(\nu^2 + 3)}{6l\nu G_3} , \qquad (3.5)$$

see [7, 21, 25] and also [8, 26] for related results.

Coming back to (3.4), for $\nu^2 > 1$ one has spacelike stretched black holes, for $\nu^2 < 1$ they are spacelike squashed, while for $\nu^2 = 1$ one recovers the BTZ solution in a rotating frame. We again stress that in order to avoid CTCs in Lorentzian signature, the squashed solutions are typically neglected. On the other hand we now turn to the thermodynamics analysis in Euclidean signature, where we can consider the full parameter space for $\nu^2$.

## 3.2 Standard thermodynamics

Here we review the standard thermodynamic properties of the black holes solutions, as previously analysed in [11, 19, 42]. We emphasize the fact that both inner and outer horizons give rise to a well-defined set of chemical potentials and conservation laws, see [31–34, 43].

We start with the conserved asymptotic charges, mass and angular momentum, computed via the ADT procedure, [44, 45],

$$M = \frac{\nu^2 + 3}{24G_3\nu}\left(\nu(r_+ + r_-) - \sqrt{r_+ r_-(3 + \nu^2)}\right) , \qquad (3.6)$$

$$J = \frac{(\nu^2 + 3)\nu l}{96G_3}\left[\left((r_+ + r_-) - \frac{1}{\nu}\sqrt{r_+ r_-(3 + \nu^2)}\right)^2 - \frac{5\nu^2 + 3}{4\nu^2}(r_+ - r_-)\right] , \qquad (3.7)$$

---

[9] These solutions are sometimes also dubbed "WAdS black holes in the canonical ensemble", see [27].

corresponding to "time translations" along $t$, i.e. the additional $U(1)$ symmetry, and "rotations" along $\theta$, i.e. the Cartan of $SL(2,\mathbb{R})$. This is the reverse of the typical holographic examples, see [6], where the additional $U(1)$ charge corresponds to rotations rather than mass. The Bekenstein-Hawking entropies of the two horizons are given by

$$S_\pm = \mp \frac{\pi l}{24 G_3 \nu} \left( r_\mp (3+\nu^2) + 4\nu \sqrt{r_+ r_- (3+\nu^2)} - 3 r_\pm (1+3\nu^2) \right) , \qquad (3.8)$$

while the inverse temperatures and angular velocities defined at the horizons are given respectively by

$$\beta_\pm = \pm \frac{4\pi l}{\nu^2 + 3} \frac{2\nu r_\pm - \sqrt{r_+ r_- (3+\nu^2)}}{r_+ - r_-} , \qquad (3.9)$$

$$\Omega_\pm = \frac{2}{2\nu l r_\pm - l \sqrt{r_+ r_- (3+\nu^2)}} . \qquad (3.10)$$

Note that these definitions are consistent with regularizing the Euclidean space such that it smoothly caps off at the given horizon. This means that for the negative subscripts the space itself is continued until $r_-$ instead of $r_+$, see also [28, 29].

With these definitions it is straightforward to verify that the first law of black hole thermodynamics is satisfied independently at each horizon,

$$\beta_\pm \delta M = \delta S_\pm + \beta_\pm \Omega_\pm \delta J , \qquad (3.11)$$

such that we can consider the on-shell actions

$$I_\pm(\beta_\pm, \Omega_\pm) = \beta_\pm M - S_\pm - \beta_\pm \Omega_\pm J , \qquad (3.12)$$

proportional to the Gibbs free energy in the grand-canonical ensemble.

We can therefore explicitly compute the on-shell actions, [10]

$$I_+ = -I_- = -\frac{\pi l}{48 G_3 (r_+ - r_-) \nu} \Big[ r_-^2 (3+\nu^2) + +8 r_- \nu \sqrt{r_- r_+ (3+\nu^2)}$$
$$- 2 r_- r_+ (9 + 11\nu^2) + r_+ (r_+ (3+\nu^2) + 8\nu \sqrt{r_- r_+ (3+\nu^2)}) \Big] . \qquad (3.13)$$

The fact that the two actions are proportional to each other was not a priori apparent and it does not happen for other black hole examples, see [28–30]. It however fits precisely with the field theory description, as we observe in due course.

### 3.3 Left and right-moving sectors

First, following [28], we define the so-called *natural*, or left- and right-moving chemical potentials, entropies, and on-shell actions,

$$\beta_{l,r} := \frac{1}{2} (\beta_+ \pm \beta_-) , \qquad \omega_{l,r} := \frac{1}{2} (\beta_+ \Omega_+ \pm \beta_- \Omega_-) ,$$
$$S_{l,r} := \frac{1}{2} (S_+ \pm S_-) , \qquad I_{l,r} := \frac{1}{2} (I_+ \pm I_-) , \qquad (3.14)$$

---

[10]Here we just write down the answer from the formula above, but of course the on-shell actions are well-defined and independently computable directly from the TMG action, (3.1), upon the addition of the appropriate Gibbobs-Hawking-York boundary term, [46, 47].

leading to an alternative version of the first law,

$$\delta I_{l,r} = \beta_{l,r}\,\delta M - \delta S_{l,r} - \omega_{l,r}\,\delta J = 0 \ , \tag{3.15}$$

such that $I_l = I_l(\beta_l, \omega_l)$ and $I_r = I_r(\beta_r, \omega_r)$.

Notice that we can come back describing the outer horizon thermodynamics by the identities

$$S_+ = S_l + S_r \ , \qquad I_+ = I_l(\beta_l, \omega_l) + I_r(\beta_r, \omega_r) \ . \tag{3.16}$$

It might at first sight seem that we have doubled the number of chemical potentials, such that going back describing the outer horizon physics with the set of variables $\beta_{l,r}, \omega_{l,r}$ is ill-defined, but it turns out that the left-moving sector is actually trivial in the grand-canonical sense. We find a constant inverse temperature and vanishing angular velocity in this sector,

$$\beta_l = \frac{4l\pi\nu}{\nu^2+3} \ , \qquad \omega_l = 0 \ , \tag{3.17}$$

and additionally, due to the identity

$$S_l = \beta_l\,M \ , \tag{3.18}$$

we find a vanishing left-moving on-shell action,

$$I_l = 0 \ , \tag{3.19}$$

which is also easy to see from (3.13) given that $I_+ + I_- = 0$. Already at this stage it is clear that the gravitational expectation for the left-moving partition function would be $Z_l = \exp(-I_l) = 1$, in agreement with the WCFT analysis, (2.22).

The right-moving sector then carries the non-trivial part of the black hole thermodynamics, with both $\beta_r$ and $\omega_r$ good thermodynamic variables given in terms of the black hole parameters as

$$\beta_r = \frac{4l\pi}{(r_+ - r_-)(\nu^2+3)}\left(\nu(r_+ + r_-) - \sqrt{r_+ r_-(3+\nu^2)}\right) \ , \qquad \omega_r = \frac{8\pi}{(r_+ - r_-)(\nu^2+3)} \ , \tag{3.20}$$

together with

$$S_r = \frac{l\pi(r_+ - r_-)(5\nu^2+3)}{24G_3\nu} = 2\pi\sqrt{\frac{c_r}{6}\left(-\frac{J}{2\pi} - \frac{M^2}{2\,k_r}\right)} \ . \tag{3.21}$$

The right-moving on-shell action simply obeys

$$I_r = I_+ = -I_- \ , \tag{3.22}$$

as evident from (3.13).

Remarkably, when expressed in terms of the fundamental variables $\beta_r, \omega_r$, the right-moving on-shell action takes the suggestive form

$$I_r(\beta_r, \omega_r) = -\frac{\pi}{12\,\omega_r}\left(c_r + \frac{3\,k_r}{\pi^2}\,\beta_r^2\right) \ , \tag{3.23}$$

with the constants

$$c_r = \frac{2\pi l(5\nu^2 + 3)}{\nu(\nu^2 + 3)\,G_3} \;, \qquad k_r = -\frac{\pi(\nu^2 + 3)}{6l\nu\,G_3} \;. \tag{3.24}$$

These are precisely the central charge and $U(1)$ level of the asymptotic Virasoro-Kac-Moody algebra, c.f. (3.5).

Perhaps most importantly, we can verify that

$$\frac{\partial I_r}{\partial \beta_r} = M \;, \qquad \frac{\partial I_r}{\partial \omega_r} = -J \;, \tag{3.25}$$

which means that we can really consider the newly defined $\beta_r$ and $\omega_r$ as the fundamental variables conjugate to $M$ and $J$, respectively.

# 4 Holographic match and quantum entropy

Given the results of the previous two sections, it is now straightforward to find an exact holographic mapping. In the grand-canonical ensemble we find, c.f. (2.23),

$$I_+ = -\frac{\pi}{12\,\omega_r}\left(c_r + \frac{3\,k_r}{\pi^2}\,\beta_r^2\right) = -\log Z_{\mathrm{WCFT}}(\bar{\tau}, \bar{\mu}) \;, \tag{4.1}$$

using the holographic identifications

$$\bar{\tau} = i\,\omega_r \;, \qquad \bar{\mu} = i\,\beta_r \;, \tag{4.2}$$

together with the gravitational central charge and Kac-Moody level in (3.5). Note that the factors of $i$ above are due to the different normalization for the chemical potentials in field theory, see (2.24), and gravity, see (3.15). As already stressed in several places, the inverse temperature of the black holes, conjugate to the conserved time-translation charge $M$, corresponds to the chemical potential for the $U(1)$ current in the warped CFT. Accordingly, the angular velocity of the black hole is part of the $SL(2,\mathbb{R})$ symmetry that relates to the modular parameter in field theory.

In the microcanonical ensemble we instead find, c.f. (2.25),

$$S_+ = \beta_l\, M + 2\pi\sqrt{\frac{c_r}{6}\left(-\frac{J}{2\pi} - \frac{M^2}{2\,k_r}\right)} = \log\rho_{\mathrm{WCFT}}(\underline{\mu}; n_r, J_r) \;, \tag{4.3}$$

upon the identifications

$$\underline{\mu} = i\,\beta_l = -\frac{2i\pi^2}{3G_3\,k_r} \;, \tag{4.4}$$

as well as

$$n_r - \frac{c_r}{24} = -\frac{J}{2\pi} \;, \qquad q_r = M \;. \tag{4.5}$$

Both of these equalities follow directly from consistency with (4.2) via the standard definitions of conjugate potentials. Again, notice that it is the mass of the black hole that plays the role of the $U(1)_{q_r}$ charge in the dual warped CFT$_2$. This, together with the

constant left-moving chemical potential $\underline{\mu}$, appears to be a special feature of the warping. The holographic identifications above are based entirely on the symmetry algebra and anomaly coefficients and are thus a rigorous consistency check for the classical existence of a WAdS/WCFT duality. [11]

However, having established the basic holographic dictionary, it is natural to ask whether a microscopic understanding of the field theory dual can provide deeper insights into the quantum entropy of the gravitational system. The answer is likely affirmative, given a well-defined unitary WCFT exists. Classical gravity captures the field theory results in the Cardy limit, which we have already explored. The exact grand-canonical partition function and the corresponding microcanonical density of states in a typical CFT are determined by modular properties, such as the modular weight. Based on the gravitational clues regarding the nature of WCFT presented here, we conclude that the grand-canonical partition function in this case should remain trivial in the left-moving sector but become a complete modular form in the right-moving sector,

$$Z_{\text{WCFT}}(\bar{\tau}, \bar{\mu}) = Z_r(\bar{\tau}, \bar{\mu}) \ , \tag{4.6}$$

and have a constant left-moving contribution to the density of states in the right-moving sector,

$$\rho_{\text{WCFT}}(\underline{\mu}; n_r, q_r) = e^{-i\underline{\mu}q_r} \int_{\mathcal{C}} \mathrm{d}\bar{\tau}\mathrm{d}\bar{\mu} \, Z_r(\bar{\tau}, \bar{\mu}) \, e^{-2\pi i \bar{\tau}\left(n_r - \frac{c_r}{24}\right) - i\bar{\mu}q_r} \ . \tag{4.7}$$

It would be interesting to gain a deeper understanding of the modular properties of $Z_r(\bar{\tau}, \bar{\mu})$ in the above formulae, as well as develop a full quantum prediction for the constant chemical potential $\underline{\mu}$. However, both tasks require a detailed microscopic perspective that goes beyond asymptotic symmetries. Achieving this is likely possible only within proper string theory embeddings. We aim to explore this topic in future work.

### Acknowledgements

We would like to thank the referee of [29] for providing interesting references that directed our interest to the present subjects. K.H. is supported in part by the Bulgarian NSF grant KP-06-N68/3.

---

[11]The relation between $\underline{\mu}$ and $k_r$ might also follow from asymptotic analysis similar to [26], but we have not shown this here. We aim to come back to this question in the future.

# A  Warped BTZ thermodynamics and holographic interpretation

In this appendix we look at the warped BTZ (WBTZ) black holes, also known as warped black holes in a quadratic ensemble, see [27]. These are a priori distinct TMG solutions that can be related to the warped AdS black holes studied in the main part of this work via a charge dependent coordinate transformation, [7, 27]. The outcome of this unusual transformation is a similarly unusual state dependent Virasoro-Kac-Moody algebra. These WBTZ solutions were further analysed by considering near-extremal limits, [27, 48], suggesting an inconsistency with the dual WCFT description.

   Having already discussed in section 2 the Cardy regimes of ordinary and warped CFTs, in the following we employ the natural variable split to the WBTZ thermodynamics without performing any coordinate transformations. We find that the thermodynamics is actually described by the standard Cardy formula for an ordinary CFT$_2$, exhibiting two different central charges $c_l \neq c_r$. This allows us to argue that the WBTZ, or the quadratic ensemble, is indeed not dual to a Warped CFT$_2$, in agreement with the outcome of the near-horizon results of [27, 48].

## A.1  WBTZ solution and thermodynamics

We start with the warped BTZ metrics, following the conventions of [27],

$$ds^2 = -N(r)^2 dt^2 + \frac{1 - 2H^2}{R(r)^2 N(r)^2} r^2 dr^2 + R(r)^2 (d\varphi + N^\varphi(r) dt)^2 \ , \tag{A.1}$$

where

$$R(r)^2 = (1 - 2H^2)r^2 - 2H^2 \frac{(r^2 - r_+^2)(r^2 - r_-^2)}{(r_+ + r_-)^2} \ , \tag{A.2}$$

$$N(r)^2 = \frac{1 - 2H^2}{R(r)^2 L^2}(r^2 - r_+^2)(r^2 - r_-^2) \ , \tag{A.3}$$

$$N^\varphi(r) = -\frac{1}{R(r)^2 L}\left((1 - 2H^2)r_+ r_- + 2H^2 \frac{(r^2 - r_+^2)(r^2 - r_-^2)}{(r_+ + r_-)^2}\right) \ . \tag{A.4}$$

Again, $r_\pm$ determine the positions of the outer and inner horizon, while $H^2$ and $L$ are related to the TMG parameters $\nu = \mu l/3$ and $l$ through the definitions

$$H^2 = -\frac{3(\nu^2 - 1)}{2(\nu^2 + 3)} \ , \qquad L = \frac{2\ell}{\sqrt{\nu^2 + 3}} \ , \tag{A.5}$$

satisfying the relation $1 - 2H^2 = \nu^2 L^2/\ell^2$. The mass and angular momentum of this black hole are given by

$$\begin{aligned} M &= \frac{(3 - 4H^2)(r_+^2 + r_-^2) - 2r_- r_+}{24 G_3 L \sqrt{1 - 2H^2}} \ , \\ J &= \frac{r_+^2 + r_-^2 - 2(3 - 4H^2)r_+ r_-}{24 G_3 L \sqrt{1 - 2H^2}} \ , \end{aligned} \tag{A.6}$$

For the outer and inner horizon the entropy is given by

$$S_\pm = \frac{\pi}{6 G_3 \sqrt{1 - 2H^2}}((3 - 4H^2)r_\pm - r_\mp) \ , \tag{A.7}$$

while the temperatures and the angular velocities are defined as

$$T_\pm = \frac{r_\pm^2 - r_\mp^2}{2\pi L r_\pm} \ , \qquad \Omega_\pm = -\frac{r_\mp}{r_\pm} \ . \tag{A.8}$$

On both horizons the first law of black hole thermodynamics is independently satisfied,

$$\beta_\pm \delta M - S_\pm - \beta_\pm \Omega_\pm \delta J = 0 \ . \tag{A.9}$$

We can thus consider the on-shell actions as

$$I_\pm(\beta_\pm, \Omega_\pm) = \frac{\pi(r_\mp + (-3 + 4H^2)r_\pm)}{12 G_3 \sqrt{1 - 2H^2}} \ . \tag{A.10}$$

## A.2  Natural variables and holographic match

With reference to the main text, we can define the natural variables as in (3.14). For the potentials we find

$$\beta_l = \omega_l = \frac{\pi L}{r_+ + r_-} \ , \qquad \beta_r = -\omega_r = \frac{\pi L}{r_+ - r_-} \ . \tag{A.11}$$

Similar conditions were found already in [30] for BTZ solutions in topological massive gravity. Indeed, these kind of solutions can be regarded as deformations of the BTZ black holes [27]. We can expect, therefore, a similar discussion as the one done in TMG. For the entropies we obtain

$$S_l = \frac{1}{6 G_3} \pi \sqrt{1 - 2H^2}(r_+ + r_-) \ , \qquad S_r = \frac{\pi \left(1 - H^2\right)\left(r_+ - r_-\right)}{3 G_3 \sqrt{1 - 2H^2}} \ . \tag{A.12}$$

Finally, the left and right moving on-shell action can be stated as follows

$$I_l = -\frac{\pi c_l}{12 \beta_l} \ , \qquad I_r = -\frac{\pi c_r}{12 \beta_r} \ , \tag{A.13}$$

where the charges are defined as

$$c_l = \frac{\pi L}{G_3} \sqrt{1 - 2H^2} \ , \qquad c_r = \frac{2\pi L(1 - H^2)}{G_3 \sqrt{1 - 2H^2}} \ . \tag{A.14}$$

Taking into account the definitions of $L$ and $H$, we find

$$c_l = \frac{4\pi l \nu}{G_3 \left(\nu^2 + 3\right)} \ , \qquad c_r = \frac{\pi l \left(5\nu^2 + 3\right)}{G_3 \nu \left(\nu^2 + 3\right)} \ , \tag{A.15}$$

clearly related to those in (3.5) for the WAdS black holes. Notice that the difference between central charges is given by

$$c_r - c_l = \frac{\pi l}{\nu \, G_3} = \frac{3\pi}{\mu \, G_3} \ , \tag{A.16}$$

which is precisely in agreement with the usual BTZ solutions in TMG and is fixed by the gravitational Chern-Simons term, [30]. Note that in this case we can also directly take the

extremal limit, $r_- = r_+$, where the right-moving sector vanishes due to $\beta_r \to \infty$, see again [30].

It is now straightforward to obtain a precise holographic match with the pure Cardy formula of a $CFT_2$, see section 2.1. In particular, we find that (2.7) is in exact agreement with (A.13) upon the identification,

$$\tau = i\,\beta_l \,, \qquad \bar{\tau} = i\,\beta_r \,. \tag{A.17}$$

Furthermore, in the microcanonical ensemble we find that the density of states (2.9) is in agreement with the WBTZ entropy $S = S_l + S_r$, (A.12), upon the identification

$$n_l - \frac{c_l}{24} = \frac{1}{2\pi}(M - J) \,, \qquad n_r - \frac{c_r}{24} = \frac{1}{2\pi}(M + J) \,. \tag{A.18}$$

These are precisely the standard holographic identifications for $BTZ/CFT_2$ correspondence, with the TMG Chern-Simons term resulting in the distinction $c_l \neq c_r$, see [30].

We have thus demonstrated that, without performing further coordinate transformations, the WBTZ thermodynamics is consistent with a standard $CFT_2$ interpretation, in contrast to the thermodynamics of WAdS black holes discussed in the main text. In the language of [27], we find that already classically there is no ambiguity between the canonical and quadratic ensembles of warped solutions.

## B   Thermodynamics in warped de Sitter space

Another interesting class of solutions of warped black holes in TMG follows from taking $\Lambda > 0$ [18, 19, 49–51]. Also for this case, we proceed by studying the thermodynamics and we employ the natural variable decomposition. The results are in close analogy with those in section 3, given the close relation between the black hole solutions.

### B.1   Warped solutions in TMG

Let us start from the solution, which is given by

$$\begin{aligned}
\frac{ds^2}{l^2} = dt^2 &+ \frac{dr^2}{(\nu^2 - 3)(r - r_+)(r + r_-)} - \left(2\nu r + \sqrt{r_+ r_-(3 - \nu^2)}\right) dt d\theta \\
&+ \frac{r}{4}\left(3(\nu^2 + 1)r + (\nu^2 - 3)(r_+ - r_-) + 4\nu\sqrt{r_+ r_-(3 - \nu^2)}\right) d\theta^2 \,. \quad \text{(B.1)}
\end{aligned}$$

In this case $\nu^2 < 3$ and both $r_+$ and $r_-$ are positive. By defining a new set of coordinates [50, 51],

$$\tilde{t} = \frac{2\nu}{3 - \nu^2}t \,, \qquad \tilde{r} = r - \frac{1}{2}(r_+ - r_-) \,, \qquad \tilde{\theta} = -\frac{2}{3 - \nu^2}\theta \tag{B.2}$$

$$\omega = \frac{2\nu^2}{3(\nu + 1)}\left(r_+ - r_- + \frac{\sqrt{r_+ r_-(3 - \nu^2)}}{\nu}\right) \,, \qquad r_h^2 = \frac{1}{4}(r_+ + r_-)^2 \,, \tag{B.3}$$

one obtains the two-parameter $(r_h, \omega)$ family of solution specified by

$$\frac{ds^2}{l^2} = \frac{4\nu^2}{(3-\nu^2)^2}d\tilde{t}^2 + \frac{d\tilde{r}^2}{(3-\nu^2)(r_h-\tilde{r})(\tilde{r}+r_h)} + \frac{2\nu}{(3-\nu^2)^2}\left(4\nu\tilde{r} + \frac{3(\nu^2+1)\omega}{\nu}\right)d\tilde{t}d\tilde{\theta}+$$
$$+ \frac{3(\nu^2+1)}{(3-\nu^2)^2}\left(\tilde{r}^2 + 2\tilde{r}\omega + \frac{(\nu^2-3)r_h^2}{3(\nu^2+1)} + \frac{3(\nu^2+1)\omega^2}{4\nu^2}\right)d\tilde{\theta}^2 \ , \quad \text{(B.4)}$$

where the black hole horizon and the cosmological horizon are located, respectively, at $-r_h$ and $r_h$, while $\omega^2 > \frac{4\nu^2 r_h^2}{3(\nu^2+1)}$ in order to avoid CTCs. These solutions can be related to the AdS one by Wick rotation of both the black hole parameters and the coordinates, c.f. (3.4) and (B.1). This analytic continuation is the reason for the close relation between the thermodynamic quantities that we discuss next.

## B.2 Standard thermodynamics

Here we set $G_3 = 1$ for a clearer comparison with literature. Corresponding to the $\partial_{\tilde{t}}$ and $\partial_{\tilde{\theta}}$ Killing vectors, the conserved charges are found to be

$$Q_{\partial_{\tilde{t}}} = \frac{(\nu^2+1)}{4\nu(3-\nu^2)}\omega \ , \qquad Q_{\partial_{\tilde{\theta}}} = \frac{3l(1+\nu^2)^2}{32\nu^3(3-\nu^2)}\omega^2 - \frac{(5\nu^2-3)l}{24\nu(3-\nu^2)}r_h^2 \ . \quad \text{(B.5)}$$

The inverse temperature and angular velocities w.r.t. the cosmological horizon are, [12]

$$\beta_c = \pi l\frac{4r_h\nu^2 + 3(1+\nu^2)\omega}{2r_h\nu^2} \ , \qquad \Omega_c = \frac{4\nu^2}{l(4r_h\nu^2 + 3\omega + 3\nu^2\omega)} \ , \quad \text{(B.6)}$$

while the entropy is given by

$$S_c = \frac{\pi l\left(3\left(\nu^2+1\right)\omega + \left(5\nu^2-3\right)r_h\right)}{6\nu\left(\nu^2-3\right)} \ . \quad \text{(B.7)}$$

All together, they satisfy the first law

$$\beta_c\delta Q_{\partial_{\tilde{t}}} = \delta S_c + \beta_c\Omega_c\delta Q_{\partial_{\tilde{\theta}}} \ . \quad \text{(B.8)}$$

For the black hole horizon one finds

$$\beta_H = \pi l\frac{4r_h\nu^2 - 3(1+\nu^2)\omega}{2r_h\nu^2} \ , \qquad \Omega_H = \frac{4\nu^2}{l(-4r_h\nu^2 + 3\omega + 3\nu^2\omega)} \ , \quad \text{(B.9)}$$

and

$$S_H = \frac{\pi l\left(3\left(\nu^2+1\right)\omega - \left(5\nu^2-3\right)r_h\right)}{6\nu\left(\nu^2-3\right)} \ , \quad \text{(B.10)}$$

again satisfying

$$\beta_H\delta Q_{\partial_{\tilde{t}}} = \delta S_H + \beta_H\Omega_H\delta Q_{\partial_{\tilde{\theta}}} \ . \quad \text{(B.11)}$$

Explicitly, the on-shell actions, defined to satisfy $\delta I_{c,H} = 0$ from the first laws above, are given by

$$I_c = -I_H = \frac{\pi l\left(5\nu^2-3\right)r_h}{12\nu\left(\nu^2-3\right)} - \frac{3\pi l\left(\nu^2+1\right)^2\omega^2}{16\nu^3\left(\nu^2-3\right)r_h} \ , \quad \text{(B.12)}$$

in analogy to (3.13).

---

[12]Contrary to [51], we label $T_c$ the Hawking temperature relative to the outer, or cosmological horizon and use $T_H$ for the temperature of the inner, or black hole, horizon.

## B.3 Left and right-moving sectors

As in (3.14), we can again split the contributions into left and right-moving sectors. We find again a constant inverse temperature and vanishing angular velocity for the left sector

$$\beta_l = 2\pi l \; , \qquad \omega_l = 0 \; , \tag{B.13}$$

and, as in (3.18), the entropy obtained to be

$$S_l = \beta_l \, Q_{\partial_{\tilde{t}}} \; , \tag{B.14}$$

As expected, the left-moving on-shell action is vanishing

$$I_l = 0 \; . \tag{B.15}$$

and the right-moving sector carries the non-trivial part of the black hole thermodynamics. For the potentials we get

$$\beta_r = \frac{3\pi l \left( \nu^2 + 1 \right) \omega}{2\nu^2 r_h} \; , \qquad \omega_r = \frac{2\pi}{r_h} \; , \tag{B.16}$$

and the entropy is given by

$$S_r = \frac{\pi l \left( 5\nu^2 - 3 \right) r_h}{6\nu \left( 3 - \nu^2 \right)} \; . \tag{B.17}$$

The left-moving on-shell action simply obeys

$$I_r = I_c = -I_H \; . \tag{B.18}$$

By defining the charges

$$c_r = \frac{2\pi l (5\nu^2 - 3)}{\nu (3 - \nu^2)} \; , \qquad k_r = -\frac{2\pi\nu}{3l (3 - \nu^2)} \; . \tag{B.19}$$

the right-moving on-shell action takes, once again, the suggestive form, c.f. (3.23),

$$I_r = -\frac{\pi}{12\,\omega_r} \left( c_r + \frac{3\,k_r}{\pi^2} \beta_r^2 \right) \; . \tag{B.20}$$

The results mimic (as expected from the analytic continuation of the metric) the behavior discussed in the WAdS black hole case. An exact holographic match would then follow the same steps as in section 4. Even though this appears straightforward, it can again be taken as a sign for the existence of a holographically dual warped $CFT_2$, which exhibits the Virasoro-Kac-Moody algebra with the constants $c_r$ and $k_r$ given above, (B.19).

It would be interesting to explore the microscopic analog of the analytic continuation between WAdS and WdS black holes and see if the corresponding WCFT is a well-defined quantum theory also in the dS case.

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
