# Peer review of "Warped AdS$_3$ black hole thermodynamics and the charged Cardy formula"

_SciPost Physics Core_

## Round 2 · Referee Report · Anonymous (Referee 1) · 2025-5-1

Report

This work tries to address some important conceptual issues associated with warped CFTs and holographic warped AdS/warped CFT correspondence. The main claim of the paper is that warped CFTs can be thought of as a 2d non-unitary chiral CFT with a crossover U(1) current. This leads to a Virasoro-Kac-Moody symmetry algebra in the right sector and a frozen left sector with a single state which, nevertheless, contributes to the entropy.

The motivation of the paper is well-founded and their proposal sounds reasonable. However, I am not satisfied that the authors have been able to support their proposal with the computations or arguments presented in the paper. More concretely, I have the following criticisms:

  1. Warped CFTs are a bit different from chiral CFTs with an internal U(1) current because the U(1) symmetry of warped CFTs is a spacetime symmetry and not an internal symmetry. This is important for deriving modular transformation behaviour of warped CFT partition function and is also responsible for appearance of an important phase factor in the transformation of the partition function. It is not clear to me with how is the interpretation of WCFTs proposed here consistent with these facts.

  2. Below equation 2.15, it is assumed that only vacuum has $h=0$. This is in contradiction with the presence of left and right moving U(1) currents which have $\bar h=0$ and $h=0$ respectively.

  3. A related question is : Is it assumed that $p=0$ for the vacuum?

  4. In appendix A, it proposed that WBTZ should be thought of as being dual to a CFT_2. This is based on being able to reproduce the entropy of WBTZ using a CFT_2 Cardy formula. However, this is achieved by ad-hoc substitution of quantities like central charges. No computation of central charges or asymptotic symmetry algebra has been presented that would substantiate their claim. Such a computation is imperative because their claim is in conflict with the analysis of arxiv: 2112.13116 where explicit boundary conditions have been found for WBTZ that show that the asymptotic symmetry algebra is the algebra of a Warped CFT in quadratic ensemble and entropy matching has been carried out with a warped Cardy formula. At the very least, it should be explained why the interpretation as a CFT_2 is more natural/ desirable in comparison to arxiv: 2112.13116.

  5. It would also be valuable if the authors could clearly compare and contrast their interpretation of warped CFTs with the standard interpretation and possibly highlight the advantages/ disadvantages of their interpretation.

I would be happy to recommend the work if above issues are addressed. However, In the current state, I cannot recommend the paper for publication.

Recommendation

Ask for major revision

---

## Editorial Decision

awaiting_resubmission